# Perceived racial discrimination, resilience, and oral health behaviours of adolescents with immigrant backgrounds

Priyanka Saluja[1], Babak Bohlouli[1], Wendy Hoglund[2], Maryam Amin[1] *

**1** Department of Dentistry, University of Alberta, Edmonton, Canada, **2** Department of Psychology, University of Alberta, Edmonton, Canada

* maryam.amin@ualberta.ca

## Abstract

### Introduction

Unmet oral health needs remain a significant issue among immigrant adolescents, often exacerbated by experiences of racial discrimination. This study aimed to examine the associations between perceived discrimination and oral health behaviours in adolescents with immigrant backgrounds and explore the potential moderating role of resilience on this association.

### Methods

Ethical approval for this cross-sectional study was obtained from the University of Alberta Research Ethics Board. Participants were 12 to 18-year-old adolescents from immigrant backgrounds. Participants were recruited through nine community organizations using a snowball sampling technique. After obtaining active parental consent and assent from the adolescent, the participants completed a questionnaire covering demographics, oral health behaviours, and perceived racial discrimination and resilience. Perceived racial discrimination and resilience were measured using validated scales. Descriptive statistics summarized variables. Logistic regression assessed associations, controlling for confounding factors. Resilience's moderating impact was analyzed via the interaction model of regression analysis.

### Results

In this cross-sectional study of 316 participants, average age of 15.3 (SD = 1.9) years, and a median age of 15 years (Inter Quartile Range-12-18), 76% reported discrimination experiences. Adjusted analysis showed that an increase of one unit in the total discrimination distress score was associated with 51% less likelihood of categorizing self-rated oral health as good (OR = 0.49, 95% CI: 0.29–0.81). The odds of brushing teeth more than twice a day, as opposed to once a day, decreased by 58% with one unit increase in the total discrimination distress score (OR = 0.42, 95% CI: 0.25–0.71). The odds of visiting the dentist for an urgent procedure instead of a regular check-up were 2.3 times higher with a unit increase in the

**Data Availability Statement:** Anonymized data is available upon request and University of Alberta ethics approval. Data are available from the University of Alberta Institutional Data Access /

Ethics Committee (contact Research Ethics Office via reoffice@ualberta.ca) for researchers who meet the criteria for access to confidential data.

**Funding:** The author(s) disclosed receipt of the following financial support for the research, authorship, and/or publication of this article: This work was supported by the University of Alberta, School of Dentistry Oral Health Community Engagement Fund (grant number- OHCEF-2022-01). The funders had no role in study design, data collection and analysis, decision to publish, or preparation of the manuscript.

**Competing interests:** The authors have declared that no competing interests exist.

total discrimination distress score (OR = 2.3: 95% CI:1.3–4.0) Resilience did not moderate the observed association.

## Conclusion

Perceived racial discrimination was associated with the pattern for dental attendance, tooth brushing frequency, and self-rated oral health. Resilience did not moderate the observed association.

## Introduction

The annual immigration rate in Canada is approximately 500,000—one of the highest per capita rates globally [1]. In 2021, Canada had a total of almost 8.3 million immigrants—nearly 23 percent of the population [2]. Statistics Canada projects that by 2041, about 52.4% of the population will be immigrants [3]. Adolescents make up a substantial population of newcomers in Canada [3]. Adolescence is an important developmental period that involves new challenges and transitions for adolescents' mental, physical, and social-emotional well-being [4]. When immigration and acculturation collide with developmental challenges, several complicated psychodynamic processes such as forming an integrated identity, or finding coping mechanisms emerge that may affect adolescents. These processes can provide a significant developmental challenge, putting adolescents at risk for mental health problems or may lead to their resiliency, and psychological growth [5].

Immigrant adolescents struggle with compounded challenges such as language barriers, and social integration as they adapt to a new culture, with different social structures and new peer relationships. Acculturation stress encompasses the mental and emotional difficulties individuals encounter while adapting to a new culture. The documented role of acculturation stress on the mental well-being of immigrant children and adolescents highlights its detrimental effects [6]. Immigration stress is the psychological strain individuals experience in response to challenges associated with adapting to a new country. Immigration stress can inhibit mental well-being and identity development in adolescence [7]. First-generation immigrants are at increased risk of emotional symptoms and psychological stress. Adolescents who are second-generation immigrants and subsequent waves of migrants are also more prone to adopting unhealthy behaviours, including substance use [8].

A scoping review on the oral health of adolescents with immigrant backgrounds in North America reported that they often experience poorer oral health compared to their non-immigrant counterparts [9]. They face limitations in accessing dental services due to language, cultural, and financial barriers [9]. An Ontario study found that immigrant adolescents in Canada were five times more prone to dental caries compared to those born in the country [10]. A study on immigrant adolescents in Spain highlighted socio-family vulnerability and deprivation among immigrant adolescents as factors contributing to the high prevalence of untreated dental caries in this group [11]. To improve oral health in adolescents with immigrant backgrounds, it is essential to study the determinants of oral health behaviours and plan intervention strategies that target these determinants of health behaviours [9]. Perceived racial discrimination is a psychosocial determinant of health, which has not been adequately investigated in relation to oral health behaviours among adolescents.

Perceived racial discrimination is characterized as the subjective perception among minority groups of unfair treatment based on race or ethnicity, often stemming from prejudice and

ethnocentrism. It can occur on individual, structural, or institutional levels [12]. A study examining the perceptions of immigrant children regarding ethnic discrimination and social exclusion in Canada unveiled that around a quarter of these children encountered discrimination from their peers, both within and outside of school, due to their unique ethnic identity [13]. According to an exploratory study of immigrants to Canada, the prevalence of racial discrimination experienced in Canada over the years 2011 to 2016 was 15.3% [14].

Perceived racial discrimination has been correlated with several mental health problems like depression, psychological distress, and anxiety [15–17]. The stress associated with experiences of discrimination may trigger physiological responses that contribute to physical health problems such as cardiovascular diseases and obesity [18,19]. A systematic review analysed the quantitative relationship between perceived racial discrimination and hypertension, finding a significant association between the two [20]. In addition, to cope with discrimination, individuals may engage in unhealthy behaviours as maladaptive coping mechanisms. Studies have reported the association between perceived racial discrimination and unhealthy behaviours like substance abuse [21]. Perceived racial discrimination has also been negatively associated with oral health. A Brazilian study on adults reported that perceived racial discrimination negatively correlated with preventive dental attendance [22]. Fear of discrimination can hinder access to both medical and dental care among immigrants [23]. In a study focussing on Aboriginal Australian adults, perceived racial discrimination was found to be negatively correlated with tooth-brushing and toothache [24,25]. In a study focused on Chinese adults in the US, there was an observed negative relationship between experiences of racial discrimination and oral health-related quality of life [26]. Similarly, a study in Canada on adolescents reported a positive correlation between perceived racial discrimination and sugar consumption frequency [27].

The theory of risk and resilience underscores the significance of identifying factors that can mitigate the negative impacts of stress and adversity on healthy development [28]. Resilience refers to individuals' capability to use external and personal strengths to foster growth when facing adversity. Factors that boost resilience during childhood and adolescence include having involved and caring caregivers, supportive family dynamics, and strong peer connections [29], religion [30], and personal characteristics such as self-regulation [31], and coping skills [32]. In the context of adversity, limited promotive or protective factors can increase an individual's risk of developing psychiatric problems, depression, anxiety, and behavioural disorders [33]. Resilience has a significant protective role in life satisfaction and general health among immigrants [34–36]. In another study on American adults, resilience moderated the association between discrimination and well-being. [37]. A Brazilian study on adults also revealed a positive relationship between resilience and how individuals rated their own oral health [38]. In another study of adolescents and adults in Nigeria, resilience played a significant role in moderating the link between anxiety symptoms and oral health problems [39].

Adolescents with immigrant backgrounds often face unmet oral health needs, as highlighted by various studies [40–42]. However, there remains a significant gap in understanding what factors influence oral health behaviours within this specific population. To address this gap, our study examined the association between perceived racial discrimination and oral health behaviours in adolescents from immigrant backgrounds. Additionally, we investigated the role of resilience as a potential moderator in this connection. It was hypothesized that the perceived racial discrimination would negatively impact oral health behaviours and resilience would moderate this association. Understanding these dynamics can help identify the risks impacting oral health practices and shed light on the factors aiding migrants in navigating challenges post-immigration.

## Methods

### Study setting

A cross-sectional study was designed. The sample size required for the study was calculated as 267 based on a confidence level of 95%, a margin of error of 6%, and a proportion of 0.5, the exact population size was unknown. In this study, participants were recruited through nine community organizations deeply involved with immigrant communities, using the snowball sampling technique. We initially approached 10 organizations in Edmonton, and nine agreed to assist in participant recruitment. These organizations host social events, activities, and religious gatherings for the immigrant population. Both parents and adolescents were introduced to the study by either the researcher or community workers during various community events organized by these groups. We began by recruiting initial participants through these community organizations and then asked them to refer others, gradually expanding our sample size. We recruited initial participants through community organizations and then we asked them to refer others, gradually expanding the sample size. The study focused on adolescents aged 12 to 18 with immigrant backgrounds who could read English. Prior to data collection, signed consent was obtained from parents, along with signed assent from the adolescent participants. The participants were explained that they could withdraw from the study at any time. To ensure accessibility, the questionnaire was available in both print and online formats. The participants were asked to answer all the questions but had the option "Do not know" in cases they had limited information about the topic of interest. This study protocol was granted ethical approval from the University of Alberta's ethics board (Ethics approval # Pro00119608).

### Data collection and procedure

The questionnaire administered to participants included four distinct sections (S1 Table). The first section included 10 questions that gathered demographic details about the adolescents and their families. The second section centered on 6 specific oral health behaviours, serving as the study's outcome or dependent variables. The third section assessed perceived racial discrimination using a 15-item validated scale and the fourth section assessed resilience using a 6-item validated scale.

### Outcome variables

In this study, outcome variables included participants' oral health behaviours and self-rated oral health. Self-rated oral health condition was assessed with a single question asking participants to rate their oral health from "very good" to "good" "fair", "poor, and "not good". Oral health behaviours were assessed by questions asking about the tooth brushing frequency (less than one, once, twice or more), sugar consumption frequency (never or less often than every day, once a day, twice a day, or more often), participant's use of dental services (within last 12 months, more than a year) the pattern for dental attendance (regular check-up, urgent/non-urgent dental problem),and smoking (yes or no).

### Independent variables

The assessment of perceived racial discrimination utilized the Adolescent Discrimination Distress Index (ADDI), a validated 15-item questionnaire [43]. This tool gauges adolescents' stress responses linked to discrimination across peer, educational, and institutional settings. Participants were asked whether they encountered specific incidents related to race or ethnicity and then rated their level of distress on a Likert scale from 1 (not at all) to 5 (extremely). The overall discrimination distress score was derived by totalling the item scores of all items and then

dividing by 15. This gives a mean discrimination distress index for each participant ranging from 1(no distress) to 5 (extreme distress). A typical item in this scale involved scenarios such as "you were given a lower grade than you deserved", followed by a question that how often they experienced it due to race or ethnicity.

Resilience was assessed using the Brief Resilience Scale (BRS), a validated tool consisting of six items [44]. Designed to measure the perceived ability to bounce back from stress, this scale includes both positively and negatively worded statements, aiming to capture an overall sense of resilience. For instance, one item states: "I tend to recover rapidly after facing challenges." Participants rated their level of agreement using a scale from 1 (strongly disagree) to 5 (strongly agree) for positively worded items (1, 3, 5), and from 1 (strongly agree) to 5 (strongly disagree) for negatively worded items (2, 4, 6). The Resilience score was computed by totalling the item scores of all items and then dividing by 6. Scores on the BRS could range from 1 (indicating low resilience) to 5 (suggesting high resilience) [44].

## Data analysis

Categorical variables were depicted as percentages, while continuous variables were summarized using means, standard deviations, and ranges when applicable. T-tests were employed for continuous variables (e.g., age), and chi-square tests for categorical variables to assess the significance of demographic variables in relation to reported racial discrimination. Based on the type of variable, different types of correlation methods were used to assess the correlation of oral health behaviour with demographics and discrimination distress score (point biserial for continuous variables and Cramer V for categorical variables). Multivariate logistic regression, employing purposeful selection of potential confounding factors, was used to examine the association between outcomes and independent variables. An interaction model of regression analysis was utilized to explore the potential moderating effect of resilience. Statistical analysis was performed using Stata-17, and statistical significance was determined by a 95% confidence interval, with p-values less than 0.05 considered significant.

## Results

### Demographics

A total of 316 participants were recruited between June 2022 and August 2023 for this study. The participants had a mean age of 15.3 years (SD = 1.9), and a median age of 15 years (Inter Quartile Range: 12–18), and 56.01% of them were female. No statistically significant age difference was observed between boys and girls (p-value <0.05). Approximately 45% of the participants were born in Canada and 62.97% possessed dental insurance. According to the adolescents, 72.78% of mothers and 71.52% of fathers had a college or university education. The racial/ethnic composition comprised Indians (31.96%), Filipino (23.42%), Chinese (15.19%), Nepalese (12.34%), African (11.39%), and Others (5.70%). Chi-square test was conducted to explore statistical proportion differences of demographic variables with and without racial discrimination. A comprehensive overview of participant demographics with and without racial discrimination is presented in Table 1.

### Oral health outcomes

Analysis of oral health behaviours revealed that more than half of the participants (57.28%) self-assessed their oral health as good. Around 62% of them brushed their teeth twice or more daily, and 73.42% consumed high-sugar foods or beverages between main meals at least once a day. Around 45.57% of participants had a dental visit in the past year. Pattern of dental visits

**Table 1. Demographic characteristics of the participants (N = 316).**

| Characteristics | n (%) | With racial discrimination (n = 241) | Without racial discrimination (n = 75) | p-value |
|---|---|---|---|---|
| Age(years)-Mean (SD) | 15.3 (1.9) | 15.5 (1.9) | 14.7 (1.9) | 0.002 |
| Median (Range) | 15 (12–18) | | | |
| Gender | | | | 0.50 |
| Female | 177 (56.01) 136 (43.04) | 136 (56.43) | 41 (54.67) | |
| Male | 3 (0.95) | 102 (42.32) | 34 (45.33) | |
| Prefer not to disclose | | 3 (1.24) | 0 (0.00) | |
| Born in Canada | | | | 0.20 |
| No | 174 (55.06) | 137 (56.85) | 37 (49.33) | |
| Yes | 142 (44.94) | 104 (43.15) | 38 (50.67) | |
| Ethnicity | | | | 0.02 |
| Indian | 101 (31.96) | 74 (30.71) | 27 (36.00) | |
| Filipino | 74 (23.42) | 59 (24.48) | 15 (20.00) | |
| Chinese | 48 (15.19) | 39 (16.18) | 9 (12.00) | |
| Nepali | 39 (12.34) | 22 (9.13) | 17 (22.67) | |
| African | 36 (11.39) | 31 (12.86) | 5 (6.67) | |
| Others | 18 (5.70) | 16 (6.64) | 2 (2.67) | |
| Living status | | | | 0.29 |
| Both parents | 268 (84.81) | 200 (82.99) | 68 (90.67) | |
| Single | 39 (12.34) | 33 (13.69) | 6 (8.00) | |
| Others | 9 (2.85) | 8 (3.32) | 1 (1.33) | |
| Father's education | | | | 0.07 |
| High school /less | 52 (17.09) | 47 (19.50) | 7 (9.34) | |
| College/university Don't know | 216 (71.52) | 165 (68.46) | 61 (81.33) | |
| | 36 (11.39) | 29 (12.03) | 7 (9.33) | |
| Mother's education | | | | 0.58 |
| High school /less | 66 (20.89) | 53 (21.99) | 13 (17.34) | |
| College/university | 230 (72.78) | 172 (71.37) | 58 (77.33) | |
| Don't know | 20 (6.33) | 16 (6.64) | 4 (5.33) | |
| Dental Coverage | | | | 0.02 |
| Yes | 199 (62.97) | 149 (61.83) | 50 (66.67) | |
| No | 97 (30.70) | 81 (33.61) | 16 (21.33) | |
| Don't know | 20 (6.33) | 11 (4.56) | 9 (12.00) | |

was reported by 293 participants with 65.53%of these visits being regular check-ups. The remaining 21 participants reported that they had never visited a dentist. The specifics of participants' oral health behaviours are outlined in Table 2.

## Perceived racial discrimination and resilience

The ADDI scale was used to gauge discrimination distress. Cronbach alpha for Peer Discrimination distress, Educational Discrimination distress, and Institutional Discrimination Distress were 0.71, 0.75, and 0.76 respectively and an overall Cronbach alpha was 0.87, this indicated that the internal consistency of the set of these items being a reliable measure to measure discrimination distress scores. The prevalence of reporting racial discrimination was approximately 76%. Notably, the mean (*SD*) scores for Peer Discrimination distress, Educational Discrimination distress, and Institutional Discrimination Distress were 1.49 (0.64), 1.42 (0.60), and 1.31 (0.49) respectively (Table 2). Overall, the data for discrimination distress shows that the majority of respondents report low levels of distress (mean values around 1.3–1.5), with moderate variability (standard deviations between 0.47–0.64). While some respondents indicated higher distress (ranges up to 4.40), the overall severity remains low. For assessing resilience, the BRS scale was used and the mean (*SD*) scores for resilience were 3.10 (0.55).

**Table 2. Dependent and independent variables (N = 316).**

| VARIABLES | n (%) |
|---|---|
| Self-rated oral health | |
| Good | 181 (57.28) |
| Not Good | 135 (42.72) |
| Toothbrushing frequency (per day) | |
| Once or Less | 120 (37.97) |
| Twice or More | 196 (62.03) |
| Sugar consumption between meals | |
| Never or less than every day | 84 (26.58) |
| Once a day or more | 232 (73.42) |
| Utilization of dental services (last year) | |
| Within 12 months | 144 (45.57) |
| Over one year | 172 (54.43) |
| Pattern for dental attendance (n = 293) | |
| Regular check-up | 192 (65.53) |
| Dental problem | 101 (34.47) |
| Smoke | |
| No | 307 (97.15) |
| Yes | 9 (2.85) |
| Prevalence of Racial discrimination | |
| Yes | 241 (76.27) |
| No | 75 (23.73) |
| | M[SD][Range] |
| Total Discrimination Distress score | 1.40 (0.47) (1.00–3.40) |
| Peer Discrimination Distress Score | 1.49 (0.64) (1.00–4.20) |
| Educational Discrimination Distress Score | 1.42 (0.60) (1.00–4.40) |
| Institutional Discrimination Distress Score | 1.31 (0.49) (1.00–4.00) |
| Resilience | 3.10 (0.55) (1.50–4.30) |

Cronbach alpha of 0.68, indicating questionable reliability of these items as a measure of resilience. (Table 2).

## Oral health outcomes and demographics: Univariate analysis

Table 3 outlines the correlation between oral health behaviours and demographic factors. Based on the type of variable, different types of correlation methods were used to assess the correlation of oral health behaviour with demographics (point biserial for continuous variables and Cramer V for categorical variables). Self-rated oral health exhibited a significant

**Table 3. Correlation of demographics and oral health behaviours.**

| Oral health behaviours | Age | Gender | Ethnicity | Living with both parents | Born in Canada | Higher Mothers' education level | Higher Fathers Education level | Having Dental Coverage |
|---|---|---|---|---|---|---|---|---|
| Self-reported Oral Health | .05 | .12 | .17 | .14* | .01 | .09 | .18* | .27* |
| Tooth brushing Frequency | .00 | .15 | .10 | .18* | -.01 | .09 | .18* | .22* |
| Sugar consumption between meals | .00 | .07 | .27* | .18* | .12* | .13 | .11 | .17* |
| Utilization of dental services (last year) | .03 | .04 | .33* | .08 | -.01 | .09 | .05 | .09 |
| Pattern for dental attendance | -.02 | .14 | .32* | .06 | -.13* | .25* | .30* | .20* |

*Significant correlation (p < 0.05).

correlation with living status, fathers' education, and dental coverage (p-value <0.05) while tooth brushing frequency was significantly associated with living status, dental coverage, and father's education (p-value <0.05). Sugar consumption between meals demonstrated a significant correlation with, being born in Canada, living status, dental coverage, and mother's education (p-value <0.05). Furthermore, the pattern for dental attendance showed significant correlations with ethnicity, being born in Canada, father's education, mother's education, and dental coverage (p-value <0.05).

## Discrimination and demographics: Univariate analysis

In the univariate analysis, perceived racial discrimination exhibited a significant association with age, ethnicity, and dental coverage. Among all participants, 241 individuals reported experiencing discrimination, with a mean (SD) age of 15.5 (1.9) years, which was 1.2 years older than those who did not report discrimination (p-value <0.05). Additionally, discrimination was significantly correlated with ethnicity, with African participants reporting the highest rate at approximately 86%, followed by Chinese participants at 80.9%, while Nepalese participants reported the lowest rate at 56% (p-value <0.05). Moreover, discrimination showed a significant correlation with dental coverage. Among the 199 participants with dental coverage, approximately 75% reported discrimination, whereas among the 97 participants without dental coverage, about 84% reported discrimination (p-value <0.05).

## Resilience and demographics: Univariate analysis

In the univariate analysis, resilience exhibited a significant association with ethnicity. The Resilience score was 0.23 more in Chinese as compared to Indians (p-value <0.05). The Resilience score was 0.17 less in Filipinos as compared to Indians (p-value <0.05). The Resilience score was 0.33 more in Filipinos as compared to Nepalese (p-value <0.05). The Resilience score was 0.40 more in Chinese as compared to Filipinos (p-value <0.05). Resilience was also significantly associated with the level of father's education. The Resilience score was 0.22 higher in participants whose fathers had college/university level of education as compared to participants whose father had a high school or less level of education (p-value <0.05). Furthermore, resilience showed a significant association with discrimination. The Resilience score was 0.26 less in participants who reported discrimination as compared to those who did not report to have experienced discrimination (p-value <0.05).

## Oral health outcomes and perceived racial discrimination: Univariate analysis

As shown in Table 4, Point biserial correlation has been reported for oral health behaviour and discrimination score. All discrimination distress scores exhibited statistically significant correlations with oral health outcomes in the univariate analysis (p-value <0.05). Educational discrimination distress scores also had significant correlations with the pattern for dental attendance, tooth brushing frequency, and self-rated oral health (p-value <0.05). Peer discrimination distress score demonstrated significant correlations with sugar consumption frequency, tooth brushing frequency and self-rated oral health (p-value <0.05). Institutional discrimination distress score demonstrated significant correlations with self-rated oral health, tooth brushing frequency, and the pattern for dental attendance (p-value <0.05).

**Table 4. Correlation of discrimination distress score and oral health behaviours.**

| Outcome variables | Peer Discrimination Distress score | Educational Discrimination Distress score | Institutional Discrimination Distress score |
|---|---|---|---|
| Self-rated Oral Health | -.13* | -.14* | -.17* |
| Tooth brushing Frequency | -.15* | -.16* | -.20* |
| Sugar consumption between meals | .12* | .07 | .09 |
| Utilization of dental services (last year) | .01 | -.07 | .00 |
| Pattern for dental attendance | .10 | .14* | .16* |

*Significant correlation (p < 0.05).

## Oral health outcomes and perceived racial discrimination: Multivariate analysis

The adjusted odds ratios of oral health behaviours in relation to discrimination distress are presented in Table 5. After accounting for dental coverage, the analysis demonstrated that self-rated oral health had a significant association with peer and institutional discrimination distress. In adjusted analysis, the odds of categorizing self-rated oral health as good decreased by 51% with one unit increase in the total discrimination distress score, after adjusting for dental coverage and mothers' education (OR = 0.49, 95% CI: 0.29–0.81). In adjusted logistic regression analyses, the chi-square for the Hosmer and Lemeshow test for the final model was 12.4, and statistically non-significant (p = 0.13) indicating that the model fit the data reasonably well. Tooth brushing frequency also exhibited a significant association with peer, educational, and institutional discrimination distress. After adjusting for the father's education and living status, the odds of brushing teeth more than twice a day decreased by 58% with one unit increase in the total discrimination distress score (OR = 0.42, 95% CI: 0.25–0.71). In adjusted logistic regression analyses, the chi-square for the Hosmer and Lemeshow test for the final model was 3.72, and statistically non-significant (p = 0.81) indicating that the model fit the data reasonably well.

**Table 5. Adjusted odds ratio of oral health behaviours and discrimination distress score: Multivariate analyses.**

| Oral health behaviors | Peer discrimination distress Odds Ratio (95% CI) | Educational discrimination distress Odds ratio (95% CI) | Institutional discrimination distress Odds ratio (95% CI) | Total discrimination distress Odds ratio (95% CI) |
|---|---|---|---|---|
| Self-rated oral health*[a] | 0.67 (0.46–0.96) * | 0.67 (0.45–1.02) | 0.49 (0.29–0.81) * | 0.49 (0.29–0.81) * |
| Tooth brushing frequency*[b] | 0.60 (0.42–0.87) * | 0.61 (0.41–0.90) * | 0.44 (0.26–0.73) * | 0.42 (0.25–0.71) * |
| Sugar consumption between meals[c] | 1.50 (1.01–2.50) | 1.30 (0.82–2.01) | 1.70 (0.89–3.22) | 1.90 (0.98–3.41) |
| Utilization of dental services[d] | 1.10 (0.70–1.56) | 0.83 (0.57–1.21) | 1.07 (0.67–1.69) | 1.01 (0.63–1.60) |
| Pattern for dental attendance*[e] | 1.50 (1.03–2.50) * | 1.60 (1.06–2.50) * | 2.20 (1.30–2.70) * | 2.30 (1.30–4.00) * |

*Significant association.

[a] adjusted for coverage (McFadden pseudo $R^2$ = 0.05).

[b] adjusted for father's education, living status (McFadden pseudo $R^2$ = 0.06).

[c] adjusted for ethnicity, Canada-born and living status (McFadden pseudo $R^2$ = 0.06).

[d] adjusted for ethnicity (McFadden pseudo $R^2$ = 0.02).

[e] adjusted for ethnicity, Canada-born, coverage, and father's education (McFadden pseudo $R^2$ = 0.15).

Sweet consumption was significantly correlated with peer discrimination distress in univariate analysis. However, after adjusting for covariates, it was not significantly associated with the discrimination scores. In the adjusted analysis, the pattern for dental attendance was significantly associated with peer, educational, and institutional discrimination distress scores. Overall, the odds of visiting the dentist for an urgent procedure rather than a regular check-up was 2.3 times higher with one unit increase in total discrimination distress (OR = 2.30: 95% CI:1.30–4.00,) after adjusting for ethnicity, Canada-born, coverage, and fathers' education. In adjusted logistic regression analyses, the chi-square for the Hosmer and Lemeshow test for the final model was 4.77, and statistically non-significant (p = 0.78), indicating that the model fit the data reasonably well.

## Oral health outcomes and resilience: Multivariate analysis

As presented in Table 6 resilience showed a positive association with some of the oral health behaviours. The odds of categorizing self-rated oral health as good increased 3.3 times with every one-unit increase in the resilience score after adjusting for dental coverage, and living status (OR = 3.30, 95% CI: 2.10–5.30). The odds of brushing teeth more than twice a day increases 2.2 times with every one unit increase in the resilience score, after adjusting for the father's education, and living status (OR = 2.20, 95% CI: 1.40–3.40). The odds of visiting a dentist for an urgent procedure rather than a dental check-up were decreased by 56%, with every one unit increase in the resilience score, after adjusting for ethnicity, Canada-born, coverage, and fathers' education. (OR = 0.44, 95% CI: 0.26–0.73).

## Moderation analysis

The moderation analyses aimed to investigate how resilience influenced the association between distress from discrimination and oral health behaviours and it was conducted using resilience both as a continuous and categorical variable. In our multivariate analysis, perceived racial discrimination showed significant associations with self-rated oral health, tooth brushing frequency, and reasons for dental visits. However, our findings did not support the expected buffering effect of resilience on the association between perceived racial discrimination and oral health behaviours. The odds ratios for the interaction between perceived racial discrimination and resilience were not significant. For self-rated oral health, the odds ratio was 0.98 (95% CI: 0.33–2.90), for the tooth-brushing frequency it was 1.5 (95% CI: 0.53–4.40), and for the pattern of dental attendance, it was 0.75 (95% CI: 0.23–2.40). These results indicate that

**Table 6. Adjusted Odds ratio of oral health behaviours and resilience: Multivariate analysis.**

| Oral health behaviors | Odds ratio (95% CI) |
|---|---|
| Self-rated oral health*[a] | 3.30 (2.10–5.30) * |
| Tooth brushing frequency*[b] | 2.20 (1.40–3.40) * |
| Sugar consumption between meals[c] | 0.66 (0.41–1.06) |
| Utilization of dental services[d] | 1.48 (0.98–2.20) |
| Pattern for dental attendance*[e] | 0.42 (0.25–0.69) * |

* Significant association.

[a] adjusted for coverage and living status (McFadden pseudo $R^2$ = 0.11) (p<0.001).

[b] adjusted for father's education, and living status (McFadden pseudo $R^2$ = 0.06) (p<0.001).

[c] adjusted for ethnicity, Canada-born and living status (McFadden pseudo $R^2$ = 0.06) (p = 0.08).

[d] adjusted for ethnicity (McFadden pseudo $R^2$ = 0.02) (p = 0.26).

[e] adjusted for ethnicity, coverage, and father's education (McFadden pseudo $R^2$ = 0.15) (p<0.001).

resilience did not demonstrate a significant moderating effect on the connection between perceived racial discrimination and oral health behaviours in our study.

## Discussion

The primary objective of this study was to explore the association between perceived racial discrimination and oral health behaviours in adolescents while exploring the potential moderating role of resilience. Among the six oral health outcomes assessed, we found that higher levels of perceived racial discrimination were associated with reduced toothbrushing frequency, poorer self-rated oral health, and specific patterns in dental attendance. However, no significant associations emerged between perceived racial discrimination and sugar consumption or the utilization of dental services in the past year. These findings offer support to the notion that elevated perceived racial discrimination corresponds to poorer oral health outcomes, following a pattern similar to how it negatively influences overall health [45]. However, our findings did not provide support for the idea that resilience acts as a moderator in the association between perceived racial discrimination and oral health behaviours.

Our study found that 65.5% of adolescents had dental check-ups in the past year, a similar prevalence of regular check-ups among adolescents has been reported in other research [27,46]. About 62% brushed their teeth twice daily, with significant links to living status, dental coverage, and father's education consistent with prior research indicating a link between adolescents' tooth brushing habits and socio-demographic factors [47,48]. Sugar consumption also correlated significantly with living status, dental coverage, and mother's education, reflecting trends reported in other studies related to parental influence and adolescents' health knowledge [49,50].

Self-rated oral health assessments are commonly employed in research when conducting clinical examinations is not feasible for participants. This measurement is reported to be broadly associated with clinical evaluations of dental health [51]. The results of our study align with several other studies reporting that increased perception of racial discrimination was associated with a decline in self-rated oral health [52,53]. Given that experiences of racial discrimination are linked to chronic stress [54] and perceived stress has been linked to poorer self-rated oral health [55], these results further underscore the intricate interplay between psychosocial factors and oral well-being.

Perceived racial discrimination has been consistently linked to reduced engagement in health-promoting behaviours [56,57]. Our study echoed this trend, indicating that individuals who reported experiencing racial discrimination were less likely to adhere to twice daily toothbrushing habits. This outcome is in alignment with another study carried out among pregnant Aboriginal women in Australia, which highlighted that high self-reported racial discrimination associated with suboptimal tooth brushing habits; perceived stress mediated this relationship [24]. Not only, has decreased adoption of healthy behaviours been observed, but perceived racial discrimination has also been associated with increased adoption of unhealthy behaviours. Several studies have demonstrated connections between perceived racial discrimination and behaviours such as smoking, alcohol consumption, and substance use [21]. The existing body of literature suggests that individuals resort to both adaptive and maladaptive health behaviours as coping strategies when faced with the stress of discrimination [56]. We included smoking, as an outcome variable in our study to assess this association, but due to the limited number of participants who reported smoking in our sample, we were not able to examine this association.

Frequency of sweet consumption was significantly associated with peer discrimination distress in univariate analysis, but after adjusting for covariates, the correlation did not remain as

significant. The link between perceived racial discrimination and dietary habits among adolescents shows mixed evidence in existing studies. While some studies demonstrate a negative association [27,58], others fail to find a clear connection [59,60]. The reason for this inconsistency is not known currently. In our sample about three quarter of the participants reported that their sugar intake between meals was once or more than once a day, but this did not correlate with racial discrimination. The lack of a significant association with perceived racial discrimination may be attributed to the influence of numerous other factors that play a more substantial role in shaping the dietary habits of adolescents [61,62].

In our study, we observed no association between the utilization of dental services in the past year and experiences of racial discrimination among adolescents. It is noteworthy that parents often bear the responsibility for ensuring their adolescents' dental attendance. Therefore, it is not surprising that we found no discernible link between adolescents' perceived experiences of racial discrimination and their utilization of dental services. Notably, existing research consistently indicates a negative association between caregivers' encounters with racial discrimination and the utilization of healthcare services for their children [63]. The complex relationship between perceived racial discrimination and healthcare utilization underscores the necessity for tailored interventions that should address not only the individual experiences of caregivers but also the systemic factors that sustain health disparities. Such insights contribute to a more informed approach to healthcare policy, aiming to ensure equitable healthcare access and outcomes for all children, irrespective of their caregivers' experiences with racial discrimination.

The pattern of dental attendance in our study exhibited a significant association with perceived racial discrimination. This finding contradicts the results of another study conducted with adolescents, which failed to report a similar association [64]. However, our findings align with previous research on general health. A US study reported that individuals who perceive racial discrimination are less likely to receive preventive health services [65]. Demographically, the presence of dental insurance showed a significant association with their pattern of dental attendance. Participants with dental insurance were found to be more likely to visit the dentist for routine procedures. This observation aligns with previous studies that highlight the lack of insurance as a substantial barrier to accessing healthcare services [66].

While studies on racial discrimination and health have predominantly focused on risk factors, limited attention has been directed towards protective factors. In our research, we explored the potential moderating influence of resilience on the connection between perceived racial discrimination and oral health outcomes. However, our findings did not reveal any evidence of a moderating effect. The existing literature on the moderating and mediating role of resilience in the relationship between racial discrimination and health outcomes presents inconsistent findings. Some studies demonstrate these effects, such as one noting that cultural resilience mediated the adverse impact of racial discrimination on stress [67]. In a Canadian study, resilience partially mediated the correlation between perceived racial discrimination and psychosomatic symptoms [68]. Similarly, resilience was reported as a partial mediator in the association between perceived racial discrimination and oral health-related quality of life among adult Chinese immigrants [26]. Conversely, other studies have not found evidence of this moderating effect [69,70]. However, it should be noted that the studies reporting the moderating effect of resilience were conducted with the adult population. Moreover, the lack of significant results in our study might be attributed to the utilization of a brief resilience assessment, which potentially did not fully capture the intricate facets of this concept. To enhance the understanding of such relationships, future investigations could consider employing more comprehensive and detailed measures of resilience [71]. The cross-sectional design may also limit the exploration of moderation/mediation effects [72].

Even though resilience didn't show a moderating effect in the relationship between perceived racial discrimination and oral health behaviour, our study revealed a positive association between resilience and oral health outcomes. This aligns with findings from other research that also highlight positive associations between resilience and health outcomes [73,74]. According to a another study, children who were bullied or faced negative emotions were less resilient [75]. Similarly, in our study, we found that adolescents who reported experiencing racial discrimination were less likely to be resilient. The findings from the literature suggest that resilient children possess emotional, social, and behavioural abilities that enable them to effectively handle life's difficulties [74]. Therefore, emphasis should be laid on fostering resilience in children's development. Resilience training programs have emerged as a valuable resource that can contribute to the cultivation of resilience attributes in the younger generation [76]. These initiatives are based on cognitive-behavioural therapy and Mindfulness-based interventions that employ a diverse range of methodologies designed to effectively instil resilience traits in children like an open discussion, role plays, practical exercises, and psychoeducation elements [77].

About three-quarters of the participants reported experiencing some discrimination because of their race. This prevalence aligns closely with findings from other studies concerning perceived racial discrimination among immigrant populations [78,79]. Given its significant impact on adolescent health, as evidenced by our findings and supported by various studies, it is crucial for community organizations and authorities to promptly implement proactive measures. The association between racial discrimination and adverse oral health outcomes highlights the broader impact of discrimination on overall well-being. This connection highlights the urgent need for comprehensive strategies that address the root causes of discrimination and its far-reaching effects. Community organizations should lead the charge by launching targeted awareness campaigns that educate the public on the harmful effects of racial discrimination. These campaigns should aim to foster environments that celebrate diversity, promote inclusivity, and actively challenge racial biases. Through education and engagement, these organizations can empower individuals and communities to recognize and combat discrimination in all its forms. Meanwhile, authorities have a critical role in implementing and rigorously enforcing anti-discrimination policies, particularly within schools and healthcare institutions. These policies must ensure that all adolescents, regardless of race or ethnicity, have equitable access to the resources and support they need. This includes creating safe spaces where adolescents can voice their experiences and receive the necessary interventions to mitigate the impact of discrimination on their health.

Our study has some limitations that need to be acknowledged. First, the cross-sectional nature of our data collection introduces constraints on our ability to explore the sequence of events and make causal inferences. Additionally, the reliance on a self-reported questionnaire for gathering data on most variables introduces the possibility of recall and desirability biases influencing the accuracy of responses. Furthermore, relying solely on self-reported data for evaluating oral health could introduce bias, and the absence of clinical measurements for oral health parameters is noteworthy. To enhance the precision of data collection in future research, integrating monitoring tools like toothbrushing and dietary charts could offer more accurate records of toothbrushing and sugar intake frequencies. In addition, including robust clinical measures, encompassing dental caries, periodontal conditions, and other pertinent variables, would provide researchers with a more comprehensive picture of oral health.

## Conclusion

Our study adds valuable evidence to the expanding pool of literature exploring the link between experiences of racial discrimination and oral health behaviours among adolescents.

While perceived racial discrimination was negatively associated with self-rated oral health, toothbrushing frequency, and the pattern for dental attendance, no association was found with the sugar consumption frequency and utilization of dental services (last year). The moderating effect of resilience was not supported by our results. Further research is necessary to comprehensively investigate various dimensions of oral health, aiming to attain a more comprehensive understanding of how perceived racial discrimination intricately impacts the overall oral health behaviours of adolescents.

## Supporting information

**S1 Table. Research questionnaire.**
(DOCX)

## Acknowledgments

Maryam Amin is the Alberta Dental Association and College Clinical Dentistry Research Chair. Priyanka Saluja received the Community and Population Oral Health Endowed Graduate Studentship and Medical Sciences Graduate Program scholarship award.

## Author Contributions

**Conceptualization:** Priyanka Saluja, Babak Bohlouli, Wendy Hoglund, Maryam Amin.

**Formal analysis:** Priyanka Saluja, Babak Bohlouli, Wendy Hoglund, Maryam Amin.

**Funding acquisition:** Priyanka Saluja, Maryam Amin.

**Investigation:** Priyanka Saluja, Babak Bohlouli, Maryam Amin.

**Methodology:** Priyanka Saluja, Babak Bohlouli, Wendy Hoglund, Maryam Amin.

**Writing – original draft:** Priyanka Saluja.

**Writing – review & editing:** Babak Bohlouli, Wendy Hoglund, Maryam Amin.

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
