## [Decision Letter · Decision Letter 0]

13 Aug 2024

PONE-D-24-14105Perceived Racial Discrimination, Resilience, and Oral Health Behaviours of Adolescents with Immigrant BackgroundsPLOS ONE

Dear Dr. Amin,

Thank you for submitting your manuscript to PLOS ONE. After careful consideration, we feel that it has merit but does not fully meet PLOS ONE’s publication criteria as it currently stands. Therefore, we invite you to submit a revised version of the manuscript that addresses the points raised during the review process.

Please submit your revised manuscript by Sep 27 2024 11:59PM. If you will need more time than this to complete your revisions, please reply to this message or contact the journal office at plosone@plos.org. Please include the following items when submitting your revised manuscript:A rebuttal letter that responds to each point raised by the academic editor and reviewer(s). You should upload this letter as a separate file labeled 'Response to Reviewers'.A marked-up copy of your manuscript that highlights changes made to the original version. You should upload this as a separate file labeled 'Revised Manuscript with Track Changes'.An unmarked version of your revised paper without tracked changes. You should upload this as a separate file labeled 'Manuscript'.If applicable, we recommend that you deposit your laboratory protocols in protocols.io to enhance the reproducibility of your results. Protocols.io assigns your protocol its own identifier (DOI) so that it can be cited independently in the future. For instructions see: https://journals.plos.org/plosone/s/submission-guidelines#loc-laboratory-protocols. Additionally, PLOS ONE offers an option for publishing peer-reviewed Lab Protocol articles, which describe protocols hosted on protocols.io. Read more information on sharing protocols at https://plos.org/protocols?utm_medium=editorial-email&utm_source=authorletters&utm_campaign=protocols.

We look forward to receiving your revised manuscript.

Kind regards,

Ashish Wasudeo Khobragade, MD

Academic Editor

PLOS ONE

2. Please provide additional details regarding participant consent. In the ethics statement in the Methods and online submission information, please ensure that you have specified what type you obtained (for instance, written or verbal, and if verbal, how it was documented and witnessed). If the need for consent was waived by the ethics committee, please include this information.

3. For studies involving third-party data, we encourage authors to share any data specific to their analyses that they can legally distribute. PLOS recognizes, however, that authors may be using third-party data they do not have the rights to share. When third-party data cannot be publicly shared, authors must provide all information necessary for interested researchers to apply to gain access to the data. (https://journals.plos.org/plosone/s/data-availability#loc-acceptable-data-access-restrictions)

a) A description of the data set and the third-party source

b) If applicable, verification of permission to use the data set

c) Confirmation of whether the authors received any special privileges in accessing the data that other researchers would not have

d) All necessary contact information others would need to apply to gain access to the data

Additional Editor Comments:

I have a few queries after going through the manuscript.

1. The abstract is started with the objectives of the study. 

2. In the result section, it is mentioned that ‘average age of 15.3 ± 1.9 years’. What is ± 1.9? Is it a standard deviation? Please clarify.

3. ‘IQR’ full form is missing in the manuscript.

4. Mention the type of correlation determined in Tables 3 and 4. Organise the tables properly.

5. The adjusted odds ratios are calculated in the manuscript. Were all the important confounders considered for analysis? Mention the results of the Goodness of Fit test of the models. Also, mention the values of pseudo-R-squared.

6. Mention exact p values in Table 6.

Reviewers' comments:

Reviewer's Responses to Questions

**Comments to the Author**

1. Is the manuscript technically sound, and do the data support the conclusions?

Reviewer #1: Yes

Reviewer #2: Yes

2. Has the statistical analysis been performed appropriately and rigorously? 

Reviewer #1: I Don't Know

Reviewer #2: I Don't Know

3. Have the authors made all data underlying the findings in their manuscript fully available?

Reviewer #1: No

Reviewer #2: Yes

4. Is the manuscript presented in an intelligible fashion and written in standard English?

Reviewer #1: Yes

Reviewer #2: Yes

5. Review Comments to the Author

Reviewer #1: Dear Authors,

Thank you for your excellent work.

The current study is on a topic of relevance to the scope of the journal. I found the paper to be very interesting and the authors had chosen an important topic in the field of social and behavioral determinants of oral health.

well-conducted and well-written paper.

best of luck

Reviewer #2: Dear colleagues

I was honored by this review participation

Based on my limited knowledge, I have few points which my aid in highlighting better presentation of this informative research

A huge and intensive effort were evidenced throughout the manuscript, Critical aspect was explored affecting community and wellbeing

The following could be beneficial:

Abstract: Very informative abstract but conclusion is limited and very generalized

Introduction:

- Overall, the introduction is inclusive with holistic view, but more should be added to physical effect of discrimination which is briefly mentioned

- a repeated statement with different references page (5) lines (128-131):

" and physical health problems like hypertension, obesity, substance abuse, and self-reported poor health (15–17), and physical health problems like hypertension, obesity, substance abuse, and self-reported poor health (18–21)"

- For analytical research like this one, it is more suitable to state a hypothesis

Methodology:

This section in the manuscript is very short and lacking details

- There is data mentioned in the abstract is mentioned under the methodology section which should be harmonized:

* Study design is missing in the methodology section while it is mentioned in the abstract as cross-sectional study.

* It was mentioned in the abstract that " nine community organizations" while number of the organizations is not mentioned in the methodology also the data about the community organization and their locations and coverage area for each (inside Alberta or outside), how they are serving immigrants (Briefly) and how you did such communication, did you communicate other organizations, but they refused or all the organization you communicated with approved to participate.

*Convenience sampling (abstract) - snowball (methodology section), besides it should be explained more in the methodology section not to be briefed as mentioning that snowball is one of the ways to perform Convenience sampling.

- Expert opinion for content validity is missing although a validated tools was used but those tools are only part of the questionnaire and the whole questionnaire should go through content validation by field experts.

- Do participants need to complete the whole questionnaire or the can leave questions without answers (optionally), also can they withdraw at any point.

- Under the " Data Collection and procedures", only questionnaire sections were mentioned without number of questions included under each section, also the template or the stem questions is not attached to the manuscript or even added as appendix.

- Under the " Outcome Variables", only self-rated oral health condition is explained while the others were just mentioned.

Results

The order of the variables in mentioning through the tables is different and also different from the order in the methodology section, this is particularly with oral health outcomes which should be unified.

- Table 1: percentages are missing the % symbol besides the numbers as this sometimes cause confusion while it is added in the other tables. Within the context talking about table 1, it is never mentioned that this table also represents the percentages of the perceived racial discrimination along with the demographic data.

- Non-respondents' number and their percentage per variable is not mentioned

- Modalities that did not receive selection under table 2 is not totally mentioned, as under the Self-rated oral health (fair & Poor) were not included even they are not selected but they are mentioned in the methodology.

- Comments on Likert scale results are missing the resultant tendency as this tendency indicative of the severity of distress.

Discussion:

- The discussion section missing any justification or comments on the initial descriptive results and almost few about the univariate analysis, while correlations only justified.

- Regarding the important point in descriptive results is the high percentage of perceived discrimination, is it normal to find this percentage and what about reported studies addressing this issue in a matter of existence before talking about correlation with oral health and the efforts spent in management of this serious issues (in brief).

- A separate section gathering recommendations with proposed actions which should be alerting to the community organizations and even the official authorities while if applicable to share this valuable investigation with them.

6. PLOS authors have the option to publish the peer review history of their article (what does this mean?). If published, this will include your full peer review and any attached files.

Reviewer #1: No

Reviewer #2: **Yes: **Moamen A. Abdalla

---

## [Author Response · Author response to Decision Letter 0]

15 Sep 2024

We have addressed all the comments and revised manuscript accordingly.

Data avalability statement has been updated

---

## [Decision Letter · Decision Letter 1]

10 Oct 2024

PONE-D-24-14105R1Perceived Racial Discrimination, Resilience, and Oral Health Behaviours of Adolescents with Immigrant BackgroundsPLOS ONE

Dear Dr. Amin,

Thank you for submitting your manuscript to PLOS ONE. After careful consideration, we feel that it has merit but does not fully meet PLOS ONE’s publication criteria as it currently stands. Therefore, we invite you to submit a revised version of the manuscript that addresses the points raised during the review process.

Please submit your revised manuscript by Nov 24 2024 11:59PM. If you will need more time than this to complete your revisions, please reply to this message or contact the journal office at plosone@plos.org. Please include the following items when submitting your revised manuscript:A rebuttal letter that responds to each point raised by the academic editor and reviewer(s). You should upload this letter as a separate file labeled 'Response to Reviewers'.A marked-up copy of your manuscript that highlights changes made to the original version. You should upload this as a separate file labeled 'Revised Manuscript with Track Changes'.An unmarked version of your revised paper without tracked changes. You should upload this as a separate file labeled 'Manuscript'.If applicable, we recommend that you deposit your laboratory protocols in protocols.io to enhance the reproducibility of your results. Protocols.io assigns your protocol its own identifier (DOI) so that it can be cited independently in the future. For instructions see: https://journals.plos.org/plosone/s/submission-guidelines#loc-laboratory-protocols. Additionally, PLOS ONE offers an option for publishing peer-reviewed Lab Protocol articles, which describe protocols hosted on protocols.io. Read more information on sharing protocols at https://plos.org/protocols?utm_medium=editorial-email&utm_source=authorletters&utm_campaign=protocols.

We look forward to receiving your revised manuscript.

Kind regards,

Ashish Wasudeo Khobragade, MD

Academic Editor

PLOS ONE

Journal Requirements:

Additional Editor Comments:

The author revised the manuscript. However, further revision is required to consider the following points.

1. Mention the pseudo-R-squared used. e.g., Nagelkerke's R², Cox and Snell’s R² etc.

2. Include the p-value for the Hosmer Lemeshow test and the interpretation of the results thereof.

3. The % symbol is repeated in the rows in Tables 1 and 2. There is no need to repeat the % symbol if it is included in the column heading. Organize the tables properly so that they are easily understandable by the reader. Spacing between a point estimate and round brackets is missing, and it is also missing in other tables.

4. Mention the exact p-values for Table 1. Mention the unit for age in Table 1.

5. Table 6 shows that the three independent variables' p-values are <0.001. p-values for the other two variables are missing.

6. In Table No. 2, under the variable ‘pattern for dental attendance’, the total is 293. Recheck once again.

Reviewers' comments:

Reviewer's Responses to Questions

**Comments to the Author**

1. If the authors have adequately addressed your comments raised in a previous round of review and you feel that this manuscript is now acceptable for publication, you may indicate that here to bypass the “Comments to the Author” section, enter your conflict of interest statement in the “Confidential to Editor” section, and submit your "Accept" recommendation.

Reviewer #1: All comments have been addressed

2. Is the manuscript technically sound, and do the data support the conclusions?

Reviewer #1: (No Response)

3. Has the statistical analysis been performed appropriately and rigorously? 

Reviewer #1: (No Response)

4. Have the authors made all data underlying the findings in their manuscript fully available?

Reviewer #1: (No Response)

5. Is the manuscript presented in an intelligible fashion and written in standard English?

Reviewer #1: (No Response)

6. Review Comments to the Author

Reviewer #1: (No Response)

7. PLOS authors have the option to publish the peer review history of their article (what does this mean?). If published, this will include your full peer review and any attached files.

Reviewer #1: No

---

## [Author Response · Author response to Decision Letter 1]

22 Oct 2024

We have addressed all the comments and revised manuscript accordingly.

---

## [Editor Report · Decision Letter 2]

24 Oct 2024

Perceived Racial Discrimination, Resilience, and Oral Health Behaviours of Adolescents with Immigrant Backgrounds

PONE-D-24-14105R2

Dear Dr. Amin,

We’re pleased to inform you that your manuscript has been judged scientifically suitable for publication and will be formally accepted for publication once it meets all outstanding technical requirements.

Kind regards,

Ashish Wasudeo Khobragade, MD

Academic Editor

PLOS ONE
---

## [Editor Report · Acceptance letter]

29 Oct 2024

PONE-D-24-14105R2 

PLOS ONE

Dear Dr. Amin, 

I'm pleased to inform you that your manuscript has been deemed suitable for publication in PLOS ONE. Congratulations! Your manuscript is now being handed over to our production team.

Kind regards, 

on behalf of

Dr. Ashish Wasudeo Khobragade 

Academic Editor

PLOS ONE